# Mechanical homeostasis of liver sinusoid is involved in the initiation and termination of liver regeneration

Jun Ishikawa[1,2,4], Makoto Takeo [1,4], Ayako Iwadate[2], Junko Koya[2], Miho Kihira [2], Masamitsu Oshima[1,3], Yuki Suzuki[2], Kazushi Taniguchi[2], Ayaka Kobayashi[2] & Takashi Tsuji[1,2✉]

Organogenesis and regeneration are fundamental for developmental progress and are associated with morphogenesis, size control and functional properties for whole-body homeostasis. The liver plays an essential role in maintaining homeostasis of the entire body through various functions, including metabolic functions, detoxification, and production of bile, via the three-dimensional spatial arrangement of hepatic lobules and has high regenerative capacity. The regeneration occurs as hypertrophy, which strictly controls the size and lobule structure. In this study, we established a three-dimensional sinusoidal network analysis method and determined valuable parameters after partial hepatectomy by comparison to the static phase of the liver. We found that mechanical homeostasis, which is crucial for organ morphogenesis and functions in various phenomena, plays essential roles in liver regeneration for both initiation and termination of liver regeneration, which is regulated by cytokine networks. Mechanical homeostasis plays critical roles in the initiation and termination of organogenesis, tissue repair and organ regeneration in coordination with cytokine networks.

[1] Laboratory for Organ Regeneration, RIKEN Center for Developmental Biology (CDB) and RIKEN Center for Biosystems Dynamics Research (BDR), Kobe, Hyogo, Japan. [2] Department of Biological Science and Technology, Graduate School of Industrial Science and Technology, Tokyo University of Science, Noda, Chiba, Japan. [3] Department of Stomatognathic Function and Occlusal Reconstruction, Graduate School of Biomedical Sciences, Tokushima University, Tokushima, Japan. [4]These authors contributed equally: Jun Ishikawa, Makoto Takeo. ✉email: takashi.tsuji@riken.jp

Organogenesis occurs only during embryonic development through complex processes involving the formation of organ germ, which arises from organ-inductive potential stem cells, and subsequent morphogenesis, resulting in the unique morphology and function of individual organs[1]. After birth, as well as after disease and injury, organs are maintained by cell turnover, tissue repair, and organ regeneration under homeostatic conditions. Regenerative phenomena, such as whole-body regeneration in planarians and limb and tail regeneration in urodele amphibians, are widely distributed in nature, although the regenerative capacity varies among species[2,3]. During regeneration, undifferentiated cells, including stem cell-derived and/or progenitor-derived cells, traditionally referred to as blastema cells, accumulate at the amputation area, and growth and differentiation of these undifferentiated cells leads to restoration of the missing part. In mammals, the regeneration capacity is very restricted, and only tissue repair occurs in most organs following disease and injury. These differences in regenerative capacity among species are known to depend on the distribution of the wide variety of stem cells with different potentials, comprising pluripotent, organ-inductive and tissue stem cells[4]. Although the molecular mechanisms, such as gene expression networks and cytokine cascades, have been elucidated in both organogenesis and regeneration in each species, it remains unknown how the events, including physical and molecular mechanisms, are induced and completed to maintain the balance of homeostasis.

The liver plays an essential role in maintaining homeostasis of the entire body through its functions, including the storage and metabolism of substances, detoxification, and production of bile, through the three-dimensional spatial arrangement of hepatic lobules, which are cord-like structures consisting of hepatocytes and sinusoids[5,6]. In contrast to other organs with specific shapes and morphologies, the mass/volume of the hepatic lobule is essential for liver function but not organ morphology. The liver is also well known to have a high regenerative capacity among mammalian organs[7]. Mammalian liver organ regeneration is known to be a form of compensatory hypertrophy, in which the volume of an organ increases due to enlargement (hypertrophy) and proliferation (hyperplasia) of its component cells[8]. The cellular and molecular mechanisms underlying compensatory hypertrophy, which is also observed in the kidney[9], and regeneration in amphibians are clearly different because the liver is a functional organ unit that receives blood from the portal vein and hepatic artery and passes blood to the suprahepatic inferior vena cava (SHIVC). Past studies have revealed that liver regeneration, i.e., compensatory hypertrophy, is complexed process involving multiple types of cells, such as endothelial cells as well as bone marrow cells, and tightly regulated by the cytokine network, including hepatocyte growth factor (HGF), epidermal growth factor (EGF), Tumour necrosis factor (TNF), vascular endothelial growth factor (VEGF), Hedgehog, Wnt/β-catenin, and transforming growth factor-beta (TGFβ)[7,10–13]. Despite multiple studies regarding the molecular mechanism of liver regeneration, in which large haemodynamic alterations occur during liver regeneration, how the liver senses its situation and the triggers for the initiation and termination of regeneration remain largely unknown. Knowledge of these aspects is expected to elucidate the sensing mechanisms associated with successive molecular networks.

Mechanical homeostasis, including gravity, shear stress, osmotic pressure, and tension, is crucial for organ and tissue development, including the establishment of morphological and functional properties[14,15]. Organ morphology and function, which are controlled by the traction force balance in cells, are essential for maintaining organ homeostasis[16]. Cells sense physical stimuli, including frictional force (shear stress) and tension

traction force, from the surrounding environment via morphological changes in cell shapes, including changes in the cytoskeleton and membrane, and respond to mechanical-biological stimuli through mechanotransducers[17]. In the vascular structure, the various physical forces generated by blood flow, including shear stress and traction force on the surface of endothelial cells, are known to play important roles in the specific morphological and physical properties of blood vessels[18]. The extension of vascular scaffolds under high blood pressure is known to play an important roles in the specific morphological and physical properties of blood vessel structures[19]. Endothelial cells have several receptors, including P2X4 and Piezo1, as physical force sensors for vascular development and diacylglycerol-sensitive canonical transient receptor potential (TRPC) subunits for sensing membrane stretching, along with G protein-coupled receptors (GPCRs) for sensing these mechanical stresses[20–22]. Based on these observations, it is hypothesized that mechanical sensors of blood vessels in the liver would have key roles as the triggering and termination signals associated with the successive cytokine networks during liver regeneration.

In this study, we identified the fluctuating parameters of the liver sinusoidal network, which were examined by a three-dimensional structural analysis of the network, under liver regeneration by using partial hepatectomy. The operation models related to these physical fluctuations in sinusoidal networks included a bypass vessel to inhibit the sinusoidal blood flow increment after partial hepatectomy, and small-liver transplantation to a large animal to increase blood flow to the transplanted liver; these models indicated that mechanical homeostasis, including tension and shear stress, is essential for the triggering signals for liver regeneration. Erk phosphorylation in both human umbilical cord endothelial cells (HUVECs) and endothelial cells isolated from the liver was observed after shear stress loading and was inhibited by a GPCR inhibitor, suramin. The production of TGFβ1, which is known to inhibit hepatocyte proliferation, was also regulated by haemodynamics in the early liver regeneration process. These results indicated that mechanical homeostatic signalling, including shear stress and tension, to sinusoidal endothelial cells prior to HGF production plays essential roles in the initiation and termination of liver organ regeneration.

## Results

**The sinusoidal network in the hepatic lobule shows stable structural parameters**. The liver is divided into five lobes, i.e., the left lobe (LL), left median lobe (LML), right median lobe (RML), right lobe (RL), and caudate lobe (CL). These lobes are mainly composed of hepatocytes and blood vessels (Fig. 1a, b). In each hepatic lobule, blood flows from the portal vein to the central vein through the capillary called the sinusoid (Fig. 1b). Hepatocytes metabolize nutrients and synthesize proteins, which is efficiently achieved by blood flow through sinusoids. Because the function of the liver is dependent on the hepatic lobules and sinusoids, we aimed to visualize and analyse the vascular structure of the hepatic lobule. To visualize the three-dimensional (3D) liver sinusoidal network, we perfused FITC-conjugated gelatine from the left ventricle of mice and obtained confocal images of the sinusoidal area (Fig. 1c and Supplementary movie 1). We generated a vascular linear image of the hepatic lobule, which allowed us to analyse the location, positional relationship, and order and angle of branching of each sinusoid (Fig. 1d and Supplementary movie 2). We analysed the branching angles and diameter of the sinusoid at each branching point (Fig. 1e). We marked a branching point as spot A and set spots B and C immediately after branching. This procedure was repeated, and all branching points were numbered as A1, A2, A3, etc. We linked the

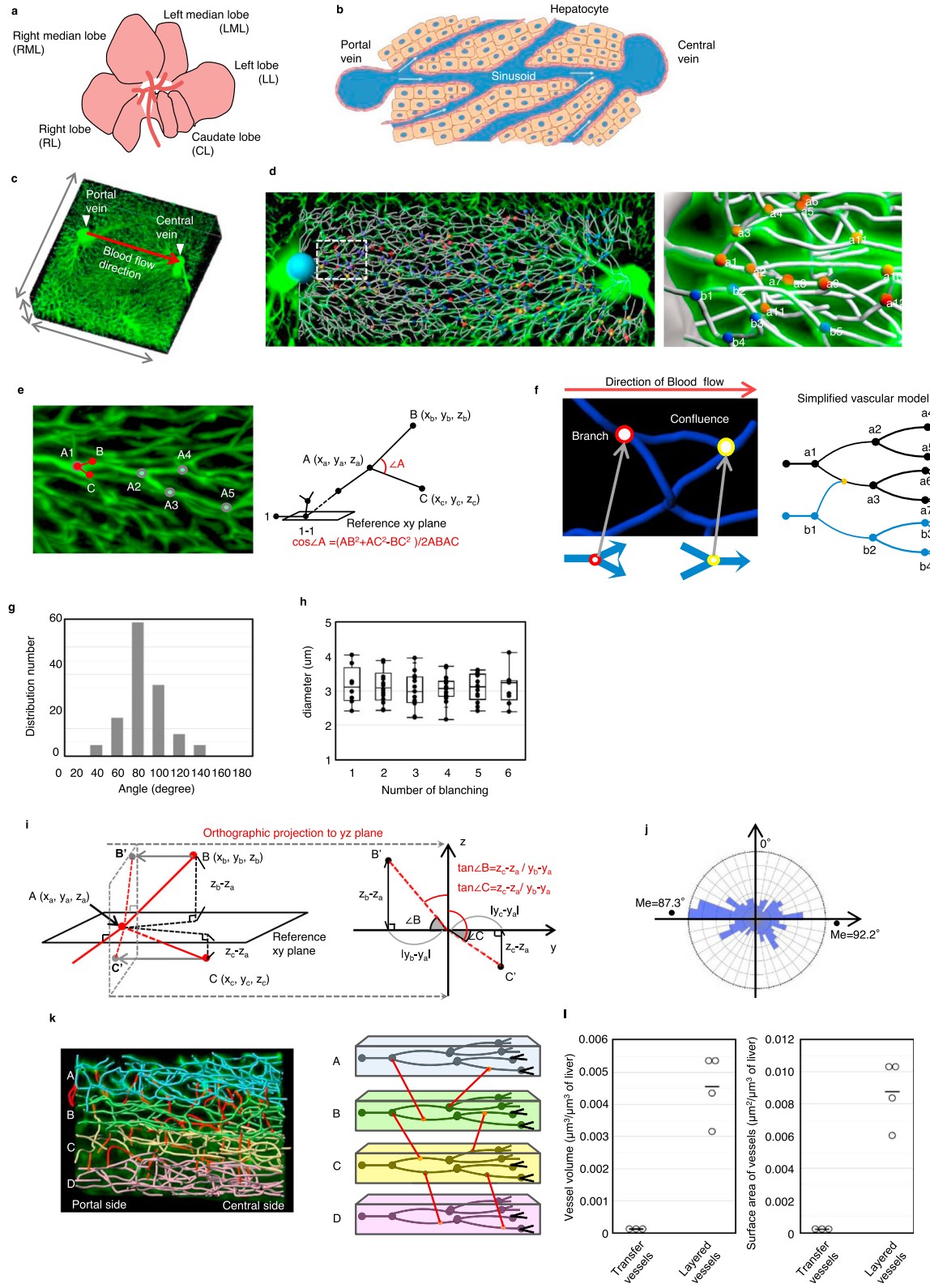

three-dimensional coordinate system to these spots and calculated the branching angle ∠A using the following formula.

$$\angle A = \arccos\angle A = arc\left(\frac{AB^2 + AC^2 - BC^2}{2AB \times AC}\right)$$

In this analysis, we considered the branching point, where the branch angle is less than 90° and two inflows join together, as the

confluent point (Fig. 1f). We found that the median branching angle was 75.75° (Fig. 1g). The sinusoidal diameter also showed a constant value of approximately 3 μm regardless of the number of branches (Fig. 1h). To analyse the rotation angle of branched vessels, we set the XY plane that contains vessels before branching as a reference plane and converted the branched vessels as an orthographic projection to the YZ plane to obtain the 2D image (Fig. 1i). We found that the value was distributed at

**Fig. 1 The sinusoidal network in the hepatic lobule shows stable structural parameters. a** Schematic of liver lobes composed of the left lobe (LL), left median lobe (LML), right median lobe (RML), right lobe (RL), and caudate lobe (CL). **b** The sinusoidal area of the hepatic lobule has a hepatic cord structure in which hepatocytes are arranged linearly along sinusoidal endothelial cells. **c** 3D image of the sinusoidal area. Blood flows to the central vein from the portal vein, and we tracked the sinusoid from the portal area to the central vein. **d** Spotting of each branching point (left) and identification of each spot (right) associating three-dimensional coordinates. **e** Measurement of branching angles and sinusoidal diameter. We labelled the new spots as B and C (left) and calculated the angle A with three-dimensional coordinates (right). **f** Schematic of two branch patterns. The red spot has a single inflow port and double outflow ports, and the yellow spot has a double inflow port and single outflow ports (left). We analysed the branches in order from the branching root (right). **g** Distribution of angles of branched vessels ($n = 105$ branching points). **h** Relation of vessel diameter and branching hierarchy ($n = 8, 14, 13, 16, 13, 9$ vessels at 1st, 2nd, 3rd, 4th, 5th, 6th branching, respectively). **i** Schematic of branching rotation angle measurement. The branched vessels were projected to the orthogonal plane (YZ plane) perpendicular to the reference XY plane. **j** Distribution of rotation angles of branched vessels ($n = 199$ branching points). **k** Layered structure of the sinusoidal network (left). Blue, green, yellow, and pink show the vessel groups with the same root. The red vessels are the transfer vessels that connected each layer (right). **l** Quantitative analysis of vascular volume and surface area in transfer vessels and layered vessels ($n = 4$ and 3 area for layered and transfer vessels, respectively).

approximately 90° against Z (Fig. 1j). Next, to understand the structural features of the sinusoidal network, we analysed the lineage of the sinusoid between the portal area and central vein. We found that sinusoids starting from the same point form a layered structure and are spatially separated from those of different origins (Supplementary movie 3). We also found that each sinusoid layer is connected to adjacent layers through another type of vessel (Fig. 1k). We defined these two types of vessels as layered vessels and transfer vessels, forming layered structures and connecting layers, respectively. In a single hepatic lobule, the volume of layered vessels was five times higher than that of transfer vessels (Fig. 1l). These findings clearly indicate that the sinusoidal network has stable structural parameters, such as branching angle and diameter, suggesting that a unique vascular network structure exists in individual organs and is maintained in a steady state under homeostatic conditions.

**Dynamic alteration of sinusoidal structure during liver regeneration.** The liver has a remarkable capacity to regenerate even after two-thirds partial hepatectomy (PH), which results in removal of the RML, LML, and LL, and the liver remnant is composed of the RL and CL (Fig. 2a). The mechanism by which the regenerating liver senses its mass and knows when to stop regenerating is unknown. Therefore, to clarify how the liver senses the need to regenerate, we focused on the blood flow velocity in the sinusoidal region because the partially hepatectomized liver has reduced volume, so the blood flow efficiency should be increased. Theoretically, before PH, the portal blood flow ($Q$) and volume ($V$) of the entire liver are expressed as the sum of each lobe (Fig. 2a). We also presumed that the portal blood is divided equally into each lobe, and the blood vessel structure of the liver is the same; thus, the blood flow efficiency, which is given as $Q/V$, is consistent in each lobe. After PH, $Q$ is unchanged despite $V$ being reduced. Therefore, the following relation between $Q$ and $V$ is given taking the inflow volume of the RL as $Q'_4$ and that of the CL as $Q'_5$.

$$Q = Q'_4 + Q'_5 \ (Q_4 < Q'_4, \ Q_5 < Q'_5)$$

$$1/3 V = V_4 + V_5$$

According to the above formula, the blood flow efficiency in each lobe ($Q'_4/V_4$ and $Q'_5/V_5$) is increased after PH. After regeneration, the total liver volume is recovered, and the blood flow efficiency in each lobe is defined as below using the regenerated volumes of the RL ($V'_4$) and CL ($V'_5$).

$$Q'_4/V'_4, \ Q'_5/V'_5$$

The liver rapidly grew to the same mass, but not shape, as the original liver by one week after PH (Fig. 2b). During regeneration, vacuolization of hepatocytes due to the accumulation of lipid droplets and subsequent swelling were observed to peak at 24 h after PH (Fig. 2c). At the same time, we also observed alteration of the sinusoidal structure, which returned to a virtually steady state by 120 h after PH (Fig. 2c and Supplementary Fig. 1). Consistent with this structural observation, 3D analysis and subsequent quantification revealed that both vascular volume and surface area increased, peaking at 1 day after PH, and then returned to the original level as liver regeneration progressed (Fig. 2d, e upper left, right). In clear contrast, the branching angle of the sinusoid showed a stable value of approximately 74° throughout the regeneration period (Fig. 2e, lower left). To analyse the effect of surrounding hepatocytes on the change in vascular structure, we performed detailed 3D image analysis and found that the nuclei of hepatocytes became sparse, and the density decreased from $15 \times 10^4$ cell per mm$^3$ to $9 \times 10^4$ cell per mm$^3$ by 1 day after PH (Fig. 2e, lower right). Next, we assumed that the value for the regenerated RL and CL is also recovered to the level before PH, and thus, the blood flow efficiency returns to normal. To examine our hypothesis, we performed in vitro imaging and directly measured the speed of erythrocytes in the sinusoids (Fig. 2f and Supplementary movies 4–6). As expected, we found that the blood flow speed drastically and significantly increased 3-fold at 12 h following PH and gradually decreased as the liver regenerated (Fig. 2g). Importantly, this change in blood flow rate occurred prior to the change in sinusoidal structure, suggesting that disruption of haemodynamics triggers the alteration of vessel structure and subsequent liver regeneration, which continues until the haemodynamics return to a steady state.

**Delay of liver regeneration under inhibition of the portal flow increment.** We further examined the causal relationship between the increase in sinusoidal blood flow velocity and liver regeneration. To this end, we developed a surgical rat model that can suppress the increased blood flow velocity in sinusoids by redirecting about 70% of blood flow of portal vein into the inferior vena cava by delegating the ligation of the portal vein to reduce adjusted, so that the cross-sectional area is to 1/3 of the original (Fig. 3a, b). In control animals, sinusoidal blood flow was immediately increased 1 h after PH and maintained a high level for over 2 days. In contrast, this acute response was neutralized in the bypass-PH group, and blood flow gradually increased after 12 h post-PH (Fig. 3c). Consistent with this finding, liver regeneration was delayed in the bypass-PH group compared to the control group at an early stage, but eventually caught up by 10 days after PH (Fig. 3d). This delayed regeneration might be associated with another regeneration system, such as metabolic demands or upregulation of blood flow by chronic arterial angiogenesis. To assess the proliferation status of hepatocytes, we performed BrdU incorporation analysis. In control animals,

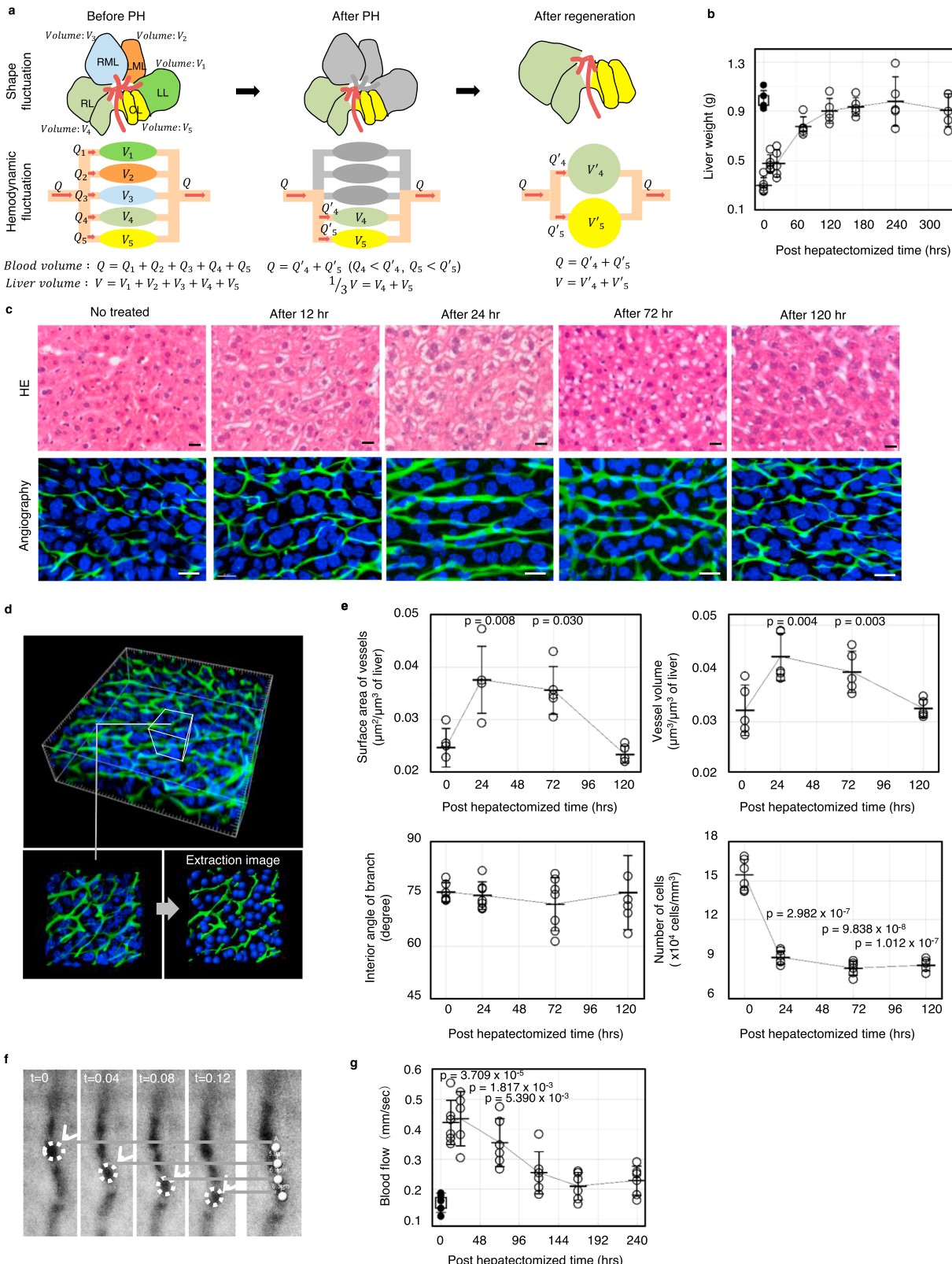

active hepatocyte proliferation, which was evidenced by the incorporation of BrdU, was observed, peaking at 1 day after PH, and ceased by 4 days after PH. On the other hand, the bypass-PH group showed poor uptake of BrdU in the early stage and maintained a low level of cell proliferation for at least 7 days (Fig. 3e). We also found that the serum level of TGFβ1, which is known as an inhibitor of hepatocyte proliferation, immediately decreased after 4 h following PH and maintained a low level for at least 12 h in the control group (Fig. 3f). In contrast, no significant decrease in TGFβ1 concentration was observed in the bypass-PH group. These results indicate that the increase in sinusoidal blood flow is vital for the suppression of TGFβ1 production and subsequent active hepatocyte proliferation, which results in rapid regeneration of the liver.

**Fig. 2 Dynamic alteration of sinusoidal structure during liver regeneration. a** Hypothesis for the effect of PH on sinusoidal flow velocity. Total blood volume ($Q$) split to each lobe as $Q_1$ (LL), $Q_2$ (LML), $Q_3$ (RML), $Q_4$ (RL), and $Q_5$ (CL). The relation of each parameter is shown in the figure before PH. After PH, the inflow volume in the RL and CL increased because $Q$ was unchanged and the sinusoidal flow velocity increased to allow blood flow. After regeneration, the $Q'_4$ and $Q'_5$ values were unchanged, but the sinusoidal flow velocity decreased because the liver volume increased. **b** Analysis of liver weight during regeneration after PH ($n = 5$ for each time point). The white square represents the liver weight before PH. **c** Histological analysis of the regenerating liver: HE image (top), angiographic image using FITC-gelatine (bottom, green: sinusoid loaded with loaded FITC-gelatine, blue: nucleus. Scale bar, 20 μm). **d** Image extraction for sinusoids and cellular nuclei for statistical analysis. **e** Quantitative analysis of sinusoidal structure in the regenerating liver: surface area of vessels (top left, $n = 5$ for each time point), vessel volume (top right, $n = 5$ for each time point), interior angle of branch (bottom left, $n = 6$ for 0 and 120 h and $n = 7$ for 24 and 72 h), and number of cells (bottom right, $n = 6$ area for each time point). Data are presented as the mean ± SD. The $p$-value is a comparison with 0 h. **f** Image of moving erythrocytes that were analysed using a fluorescence microscope and a high-speed camera every 0.04 s. **g** Blood flow rate in sinusoids during liver regeneration. The white square represents the sinusoidal flow rate before PH ($n = 5$ for 0 and 24 h and $n = 6$ for 12, 72, 120, 168, and 240 h). Data are presented as the mean ± SD. The $p$-value is a comparison with 0 h. The white square represents the liver weight before PH.

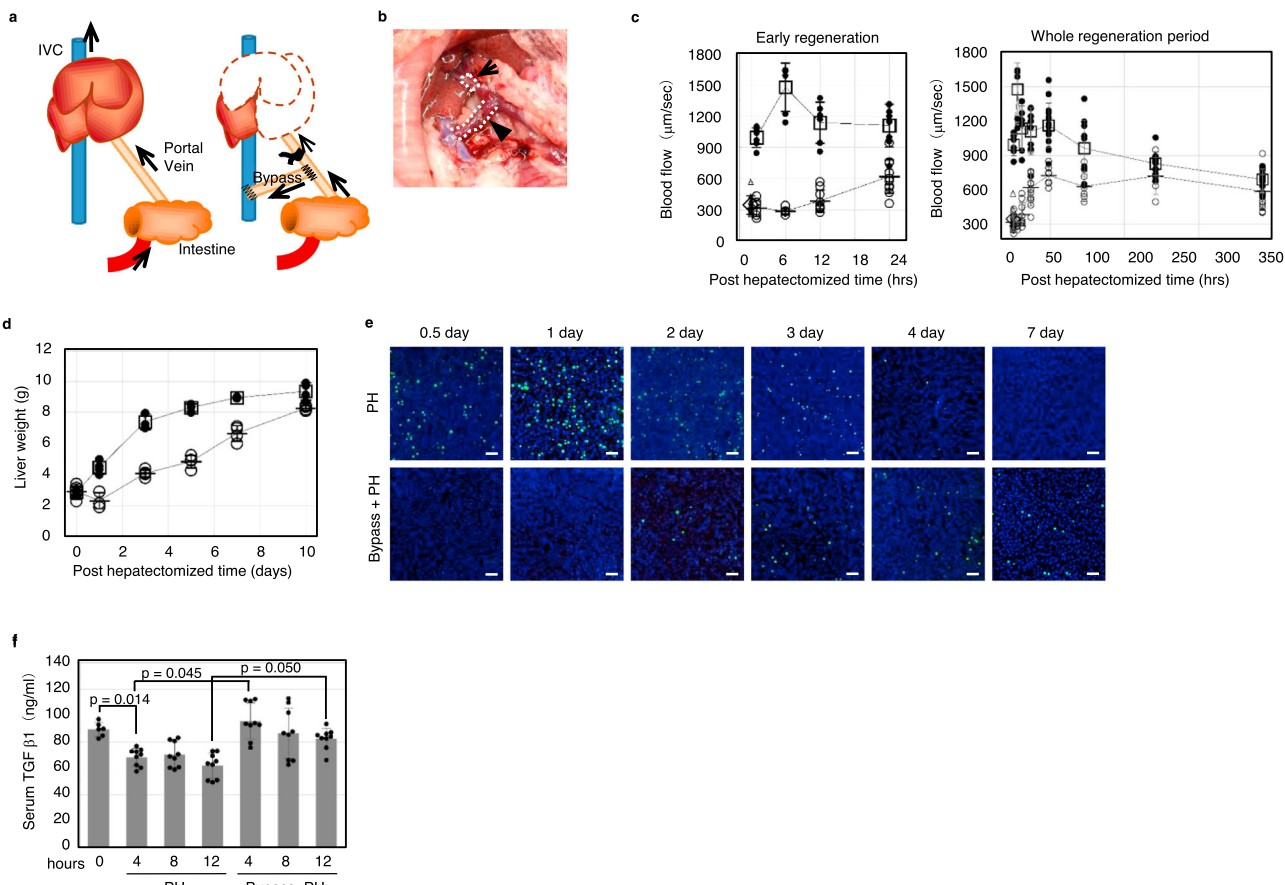

**Fig. 3 Delay of liver regeneration under inhibition of the portal flow increment. a** Schematic of bypass surgery. The portal vein was connected to the inferior vena cava and constricted using a plastic cuff to restrict the portal flow into the liver. Following this treatment, the left and median lobes of the liver were removed. **b** Photograph of the bypass technique (arrow: plastic cuff, arrowhead: bypass vessel). **c** Blood flow rate of the regenerating liver in the early regeneration phase and whole regeneration period (white circle: the group with ordinary partial hepatectomy without bypass treatment, black circle: the group with partial hepatectomy with bypass treatment. The white square represents the value before PH). $n = 11, 6, 6, 6, 8, 14, 11, 11$ for 0, 1, 6, 12, 24, 48, 96, 192, 336 h. Data are presented as the mean ± SD. **d** Recovery of liver weight during liver regeneration (white circle: the group with ordinary partial hepatectomy without bypass treatment, black circle: the group with partial hepatectomy with bypass treatment). $n = 4, 3, 5, 4, 4, 4$ for 0, 1, 3, 5, 7, 10 days after treatment in shunt group and $n = 4, 4, 3, 4, 3, 4$ for 0, 1, 3, 5, 7, 10 days after treatment in hepatectomized group. Data are presented as the mean ± SD. **e** Brd-U incorporation into the regenerating liver (green: Brd-U-positive cells, blue: nucleus. Scale bar, 100 μm). **f** Serum TGFβ1 level of partially hepatectomized rats with or without bypass construction ($n = 3$ for each timepoint.). Data are presented as the mean ± SD.

## Vascular endothelial cells sense shear stress via GPCRs.

Past studies have demonstrated that vascular endothelial cells, which are distributed throughout the liver, can detect the change in shear stress in blood vessels and shift the entire liver to the regeneration phase[21]. Therefore, we analysed how vascular endothelial cells are involved in the initiation of liver regeneration.

To this end, we developed a perfusion culture system that allows us to control the flow rate of culture medium and the strength of shear stress on human umbilical vein endothelial cells

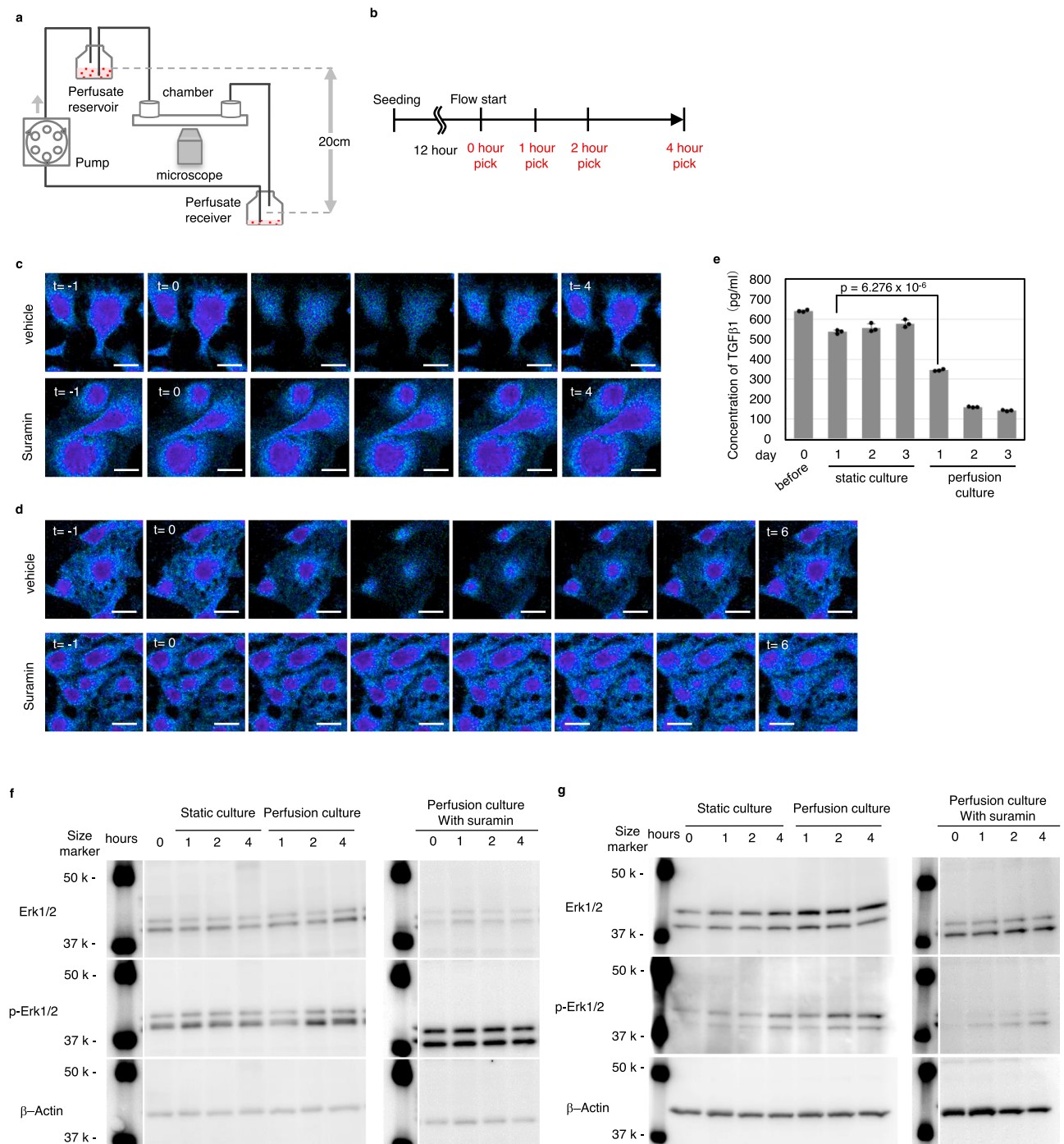

**Fig. 4 Vascular endothelial cells sense shear stress via GPCRs. a** Schematic of the shear stress loading circuit. The chamber was placed at 37 °C during culture and observation. **b** Time schedule of the shear stress test using liver sinusoidal endothelial cells. **c** Image of calcium influx under shear stress loading using HUVECs. Each image was obtained at intervals of one second. Scale bar, 10 μm. **d** Image of calcium influx under shear stress loading using LSECs. Each image was obtained at intervals of 1 s. Scale bar, 10 μm. **e** Concentration of TGFβ1 in supernatants using LSECs for 3 days under loading shear stress and static culture (*n* = 1 and 3 for before and after culture, respectively). Data are presented as the mean ± SD. **f** Western blot analysis of Erk1/2 phosphorylation under shear stress with suramin using HUVECs. **g** Western blot analysis of Erk1/2 phosphorylation under shear stress with suramin using LSECs.

(HUVECs) (Fig. 4a). When endothelial cells sense the flow, calcium ions flow into cells via cation channels on the cell membrane, and the intracellular concentration is increased[23]. We found that the intracellular calcium concentration increased immediately after shear stress loading at 15 dyne/cm$^2$, and this response was neutralized by applying suramin, an inhibitor of

GPCRs (Fig. 4c). We further determined whether liver sinusoidal endothelial cells (LSECs), obtained from intact rat liver, were capable of detecting an increase in shear stress in our in vitro model. We observed an increase in the intracellular Ca$^{2+}$ level upon loading with shear stress at 15 dyne/cm$^2$, and the effect was neutralized by applying suramin, as observed in HUVECs

(Fig. 4d). Next, we analysed whether shear stress affects the production of TGFβ1 by LSECs. In static culture conditions in which shear stress was not applied to the cells, the amount of TGFβ1 was kept constant throughout the experimental period (Fig. 4e). In clear contrast, in the perfusion culture condition, the production of TGFβ1 was drastically decreased in response to shear stress loading, indicating that LSECs are the source of TGFβ1 (Fig. 4e). We further analysed the phosphorylation status of Erk1/2, which is known to induce the proliferation of endothelial cells. Immediately after applying shear stress, the phosphorylation level of Erk1/2 gradually increased, suggesting the activation of the MAPK cascade, and this activation was suppressed by suramin in both HUVECs and LSECs (Fig. 4f, g and Supplementary Figs. 2 and 3). These results indicate that endothelial cells have the ability to sense shear stress, which in turn induces TGFβ1 production through the GPCR-MAPK axis.

**Acute liver growth is induced by an increase in portal flow.** Our observations thus far indicate that the delayed change in blood flow is involved in the control of organ size. To examine this point, we developed a small orthotopic liver transplantation (s-OLT) model in which a small liver isolated from a young rat was orthotopically transplanted into an aged large rat, and assumed that the transplanted liver regenerates similar to the PH model due to an increase in blood flow (Fig. 5a). We orthotopically transplanted donor livers isolated from 5-week-old Wistar rats into 15-week-old recipient rats. The average blood flow rate in the 5-week-old and 15-week-old rats was 11.6 ml/min and 22.4 ml/min, respectively (Fig. 5b). At 24 h after s-OLT, swelling of the small liver was observed, as seen after PH, suggesting that similar phenomena occur after both s-OLT and PH (Figs. 2c and 5c). The rate of increase in liver weight in the s-OLT model was also consistent with that in the PH model (Figs. 2b and 5d). Moreover, the average donor liver weight rapidly and significantly increased from 4.36 to 9.41 g at 10 days after transplantation, while the weight of the age-matched control liver increased slightly to 5.1 g, suggesting that rapid growth of the small liver in the s-OLT model is not natural as similar to the case of liver transplantation from baboon to human[24] (Fig. 5e). Consistent with these observations, vacuolization in the hepatocytes and chromosomal condensation were observed following both PH and s-OLT at 24 h and at 48 h after operation, respectively (Fig. 5f). In the s-OLT model, the vessel surface area and volume significantly increased by 24 h after transplantation (Fig. 5g–i). The number of cells around sinusoids also drastically decreased at 48 h after transplantation (Fig. 5j). In both the PH and s-OLT models, the expression of *TGFβ1* in LSECs was markedly decreased by 6 h after the surgical procedure, when the sinusoidal flow rate reached the maximum in the PH model (Figs. 3c and 5k). Moreover, the expression of *cyclin D1*, a cell-cycle progression marker, drastically increased in hepatocytes of both the PH model and s-OLT model, suggesting that the rapid liver growth seen in the s-OLT model occurred due to hepatocyte proliferation (Fig. 5l). These results indicate that an increase in blood flow rate results in a change in sinusoidal structure, downregulation of TGFβ1 production, and subsequent liver regeneration (Fig. 5m). Taken together, our results strongly suggest that a change in the sinusoidal blood flow rate triggers the initiation and termination of liver growth and regeneration.

## Discussion
In our current study, mechanical homeostasis indicated by fluctuation parameters, including tension and shear stress on liver sinusoidal networks, plays an essential roles in the initiation of liver regeneration. The physical fluctuations are converted to chemical signalling prior to cytokine production. It is suggested that mechanical homeostasis is involved in maintaining liver homeostasis, including the pathology and functions of both steady-state and regeneration of the liver in vivo. These findings contribute to the understanding of organ homeostasis and functions through tension homeostasis, and they indicate that mechanical homeostasis could be essential for the initiation and termination of organ regeneration and growth.

Liver organ regeneration is known to be a form of compensative hypertrophy, occurring through the enlargement (hypertrophy) of parenchymal cells, proliferation (hyperplasia) of its component cells and vascular remodelling[8]. Hepatic lobule structure improves metabolic efficiency, and the mass/volume of hepatic lobules is essential for total liver functions in vivo[25]. Previous studies, including a liver regeneration model in silico, indicated that the hepatic lobule structure ensures optimal exchange of metabolites in sinusoids[26,27]. During liver regeneration by partial hepatectomy, rapid change of hepatocyte membrane potential occurs immediately after partial hepatectomy possibly mediated by satellite cells[28,29]. Simultaneously, haemodynamic changes, such as changes in the portal pressure and physical force caused by blood flow, might be affected by sinusoidal structural parameters, such as the vascular diameter or volume, depending on the physical vasodilation force from inside the sinusoid[30,31]. Moreover, previous studies suggested the relationship between change in blood flow rate and resulting shear stress and liver regeneration[32–34]. However, the relationships between the effect on sinusoidal structure and liver regeneration remain unclear[35,36]. In the current study, we revealed the detailed structural changes in the sinusoids, including quantitative parameters, by accurately measuring the spatial arrangement of the sinusoids and found that the sinusoidal volume and the surface area drastically changed at only 6 h after the partial hepatectomy, similar to changes in parameters associated with liver volume prior to the biological reactions for liver regeneration, such as HGF expression, as previously reported[37]. Furthermore, our developed bypass model of partial hepatectomy, which suppresses the increase in blood flow and changes in sinusoidal structure, revealed that the increase in these haemodynamic changes is essential for the early stage of liver regeneration. These observations suggest that alterations in mechano-homeostasis, which is maintained by stable blood flow in sinusoids, would be the first triggers/initiators of liver regeneration.

Environmental mechanical stresses, including shear stress, osmotic pressure, and tension, are known to play an essential roles in the morphological and physical properties of tissues and organs during both embryogenesis and homeostasis after birth[14,15,38]. The traction force balance in cells also plays an important role in maintaining organ homeostasis by controlling organ morphology and function[16]. The balance among extracellular forces exerted on neighbouring cells and the traction forces generated by cells themselves is termed tensional homeostasis[39]. Among many cells and tissues, endothelial cells in vascular networks are well known to be constantly exposed to mechanical stress, including shear and tension stress on their cell membrane, caused by blood flow and/or pressure[40,41]. Previous studies reported several essential molecules related to these mechanical forces on endothelial cells[20,42]. The mechanoreceptor P2X4, which is a type of $Ca^{2+}$-ion channel on the surface of endothelial cells, followed by proteins such as calmodulin and downstream protein kinases in intracellular cascades, regulates cellular behaviour, including increased production of nitric oxide and prostacyclin, which cause vasodilatation[43]. The TRP ion channel is also known to regulate cellular functions, including endothelial permeability in actin-stress fibre formation, and is coordinated with G-protein-coupled receptors (GPCRs) as a

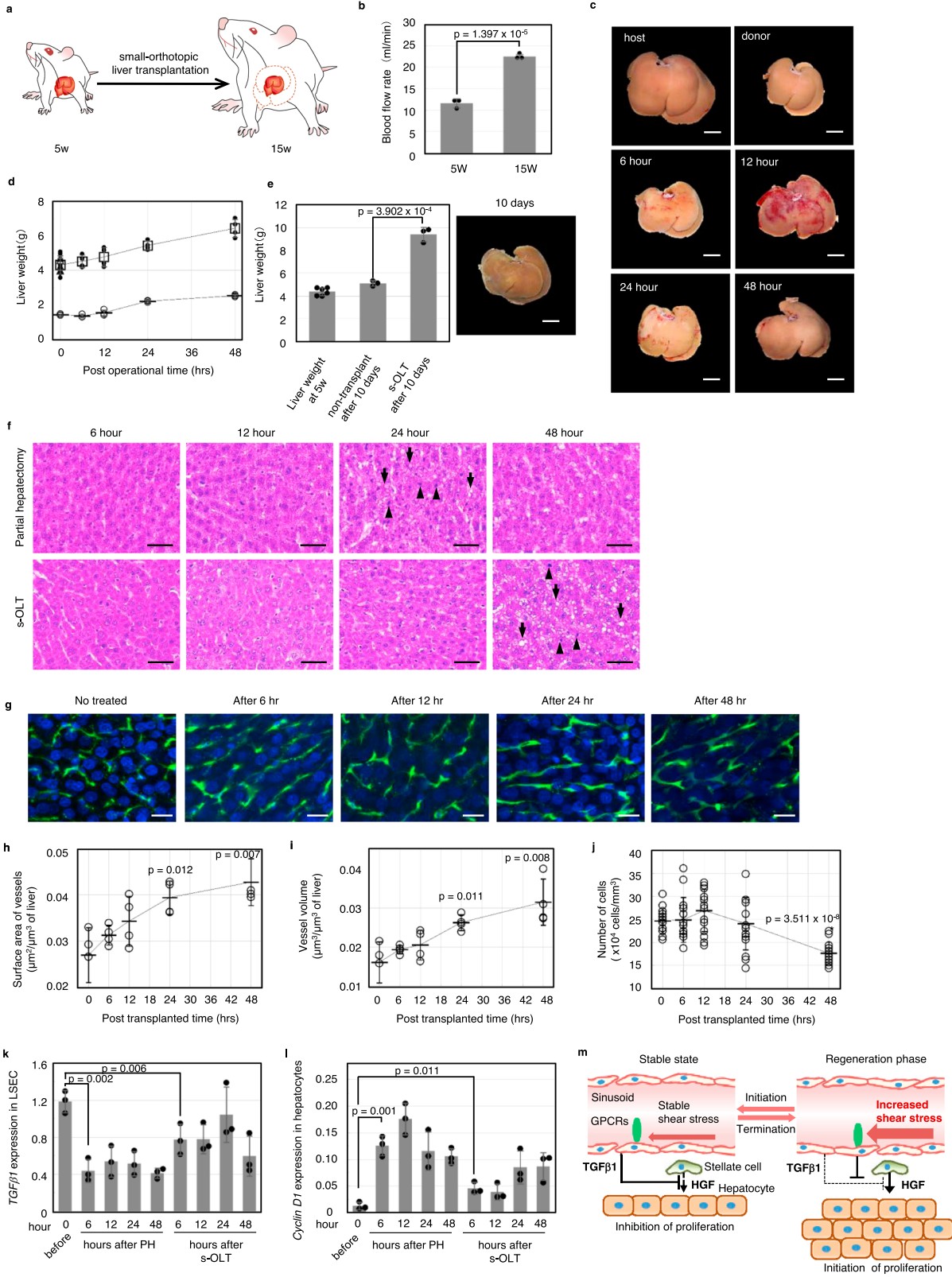

sensor for membrane stretching[42]. It is also known that several GPCRs, including angiotensin receptor type-1 and bradykinin B2 receptor, are activated by changes in the traction force balance of endothelial cells and regulate endothelial cell behaviour through conformational changes by partial rotation of the receptor unit without ligands[44–46]. In previous studies, liver regeneration was shown to be sensitive to suramin, which is an inhibitor of GPCRs[47], suggesting that mechanical homeostasis through suramin-sensitive GPCRs has an important roles in the response of liver regeneration[48–50]. In this study, we observed a delay in liver regeneration in bypass surgery to decrease the blood flow of the portal vein increment after partial hepatectomy. We also

**Fig. 5 Acute liver regeneration is induced by an increase in portal flow. a** Schematic of the small orthotopic liver transplantation (s-OLT) model. The donor liver was harvested from 5-week-old Wistar rats and transplanted to 15-week-old Wistar rats. **b** Blood flow rate in the portal vein of donors and recipients (n = 3 for each age). Data are presented as the mean ± SD. **c** Photographs of the transplanted liver over 48 h. Scale bar, 1 cm. **d** Recovery of liver weight during the early liver regeneration phase (white circle: ordinary partial hepatectomy group, black circle: s-OLT group. The white triangle represents the weight before PH. n = 3 and 4 for partial hepatectomy group and s-OLT group, respectively). Data are presented as the mean ± SD. **e** Analysis of liver weight increment after 10 days in the s-OLT model and photographs of the transplanted liver (n = 6 and 3 for 5-week-old and 10 days after treatments). Data are presented as the mean ± SD. Scale bar, 1 cm. **f** Histological analysis of the regenerating liver by HE staining (arrow: vacuolization by accumulation of lipid droplets, arrowhead: chromosomal condensation of the nucleus. Scale bar, 50 μm). **g** Angiographic image obtained using FITC-gelatine in the regenerating liver (green: FITC-gelatine loaded into sinusoids, blue: nucleus. Scale bar, 20 μm). **h–j** Quantitative analysis of sinusoidal structure in the regenerating liver for 48 h. The graphs present the surface area of vessels (**h**, n = 4 for each time point), vessel volume (**i**, n = 4 for each time point) and number of cells (**j**, n = 5 for each time point) in a constant volume. Data are presented as the mean ± SD. **k** Expression analysis of *TGFβ1* in LSECs of PH and the s-OLT model for 48 h (n = 3 for each time point). Data are presented as the mean ± SD. **l** Expression analysis of *cyclin D1* in hepatocytes of PH and the s-OLT model for 48 h (n = for each time point). Data are presented as the mean ± SD. **m** Schematic of the liver regeneration model triggered by sinusoidal flow increment by controlling *TGFβ1* expression.

observed an early-stage proliferation of parenchymal cells in the liver and downregulation of serum TGFβ1, which is known to be an inhibitor of parenchymal cell growth. We also found that $Ca^{2+}$ influx, the downregulation of TGFβ1 and Erk phosphorylation in both HUVECs and primary isolated LSECs occurred in the shear stress model in vitro and were sensitive to suramin treatment. We could not evaluate whether a similar mechanism exist in vivo situation because of the experimental limitations, such as absence of reliable method to monitor $Ca^{2+}$ influx and phosphorylation of ERK in vivo. Nevertheless, these findings strongly suggest that the balance of suramin-sensitive mechanical homeostasis in endothelial cells in the sinusoidal network in the liver plays an essential roles in liver regeneration, i.e., compensatory hypertrophy prior to cytokine networks.

During embryogenesis and organ and/or organ system regeneration, the regulation of organ size and morphology is thought to play essential roles in proper organogenesis and regeneration of the organ and/or organ system[14,15,38]. During organogenesis, tissue repair, and organ regeneration, mechanical homeostasis is considered as one of the fundamental and primitive triggers for cellular proliferation and tissue maturation prior to cytokine networks and is also essential for the termination of tissue and organ morphogenesis[38]. The mechanisms by which organs sense the timing to start and terminate organogenesis and regeneration are fundamental, but remain unanswered questions in developmental and medical biology[51–53]. Mechanical stress is thought to be a fundamental and primitive trigger not only for the initiation of cell proliferation and tissue maturation but also for the termination of regeneration to prevent over-regeneration[54,55]. In organogenesis, the cardiac loop is formed by unbalanced forces in left–right looping directionality in cardiac morphogenesis, and several mechanoreceptors mediate vascular network formation[56]. After birth, muscle or bone component cells receive mechanical force and regulate the balance between the proliferation and differentiation phases[57]. Endothelial cells also sense high blood pressure and enter vascular remodelling to reduce the pressure[20]. During liver regeneration, Hippo signalling defines an organ size through gene expression controlled via nuclear localization of the mechanical stress receptors YAP/TAZ[58–61]. These findings suggest that the balance of mechanical homeostasis, including traction force, shear stress, and other mechanical stresses, plays an essential roles in organogenesis, tissue repair, and organ regeneration. It is also well known that several cytokines, such as HGF, EGF, and inflammatory signals, mediate the liver regeneration process[13,62–64]. In this study, the change in haemodynamics following partial hepatectomy resulted in the disturbance of mechanical homeostasis, which is sensed through GPCRs and drives subsequent cytokine cascades as a trigger in the initiation phase. Furthermore, in our experiments, including the

3D-sinusoidal analysis, s-OLT model, and endothelial cell assay in vitro, these mechanical stress parameters continued to reach the equilibrium rates of those in the normal static phase and were involved in the termination process of liver regeneration for a return to equilibrium. Although further confirmation in vivo situation is necessary, these results strongly suggest that physical sensing of mechanical homeostasis is an essential system for both the primitive trigger in the initiation phase and sensing of liver size in the termination phase of liver regeneration, which is coordinated with cytokine networks.

In conclusion, we demonstrated that mechanical homeostasis in liver sinusoidal networks plays an essential roles in both the initiation and termination of liver regeneration. Our results show the impact of mechanical homeostasis in linking cytokine cascades in a wide variety of phenomena, such as organogenesis, tissue repair, and organ regeneration. Further studies on the detailed molecular mechanisms of mechanical homeostasis, including traction force balance and shear stress, which are sensed by GPCRs and other sensing molecules on endothelial cells as mechanical sensors, would provide a valuable information for understanding the roles of both systems of mechanical homeostasis and cytokine networks during organogenesis, morphogenesis, and regeneration in vivo.

## Methods

**Animals**. 5–15-weeks-old male Wistar rats and 5–8-weeks-old male C57/BL6 mice were purchased from SLC Inc. (Japan). All handling and care procedures for the rats conformed to the National Institutes of Health (NIH) guidelines for animal research, and all experimental protocols involving animals were approved by the Riken Animal Care and Use Committee (Permit Number: A2014-02-12). All efforts were made to minimize suffering.

**Analysis of vascular images**. To visualise the sinusoidal structure, we prepared a solution of fluorescein 5-isothiocyanate (FITC; Dojindo, Kumamoto, Japan) conjugated to gelatine (Sigma-Aldrich Japan), and dissolved in 1 mL dimethyl sulfoxide (DMSO; Sigma-Aldrich Japan) at pH 11. The FITC solution and 20% (w/v) gelatine solution were mixed for conjugation at 37 °C for overnight. The unconjugated FITC was removed using a NAP-25 column (GE Healthcare UK Ltd., Buckinghamshire, England). We perfused this FITC-conjugated gelatine from the left ventricle, and livers were harvested and fixed with Zamboni solution at freezing temperature. From these samples, frozen sections (300 μm thick) were prepared after fixation using 8% formalin containing Hoechst 33342 dye (1:500, Thermo Fisher, USA). The fixed sections were washed with PBS (−) and examined using laser confocal microscopy (LSM780, Carl Zeiss, Germany). These three-dimensional imaging data were analysed using Measurement, FilamentTracer, and Surface in Imaris software (Bitplane) with an x, y, and z coordinate system.

**Histochemical analysis and immunohistochemistry**. The livers were flushed with saline and fixed with 10% formalin (Mildfolm 10 N; Wako, Japan). Tissue sections (5 μm thick) were taken after paraffin embedding and staining with haematoxylin and eosin (HE). For Brd-U staining, sections were processed under 2 N HCl for 15 min at room temperature and washed three times using PBS. After blocking, sections were first stained with anti-Brd-U antibody (ab6326, Abcam,

UK) and then with FITC-conjugated goat anti-rat IgG antibody (AP183F, Merck, USA).

**Preparation of hepatocytes and LSECs**. The preparation of liver cells was performed by using the collagenase perfusion method. Abdominal incisions were performed on the rats under anaesthesia, and 18G catheters (Terumo, Japan) were inserted into the PV and inferior vena cava. The initial perfusions (100 mL; flow rate, 10–15 mL/min) were performed with the perfusion solution,which included NaCl (136.89 mM, Wako, Japan), KCl (5.37 mM, Wako, Japan), NaH$_2$PO$_4$/2H$_2$O (0.38 mM, Wako, Japan), Na$_2$HPO$_4$/12H$_2$O (0.17 mM, Wako, Japan), HEPES (10 mM, DOJINDO, Japan), glucose (5 mM, Wako, Japan), ethylene glycol tetra-acetic acid (0.5 mM, DOJINDO, Japan), NaHCO$_3$ (4.16 mM, Wako, Japan), and phenol red (0.017 mM, Sigma-Aldrich Japan, Japan). A secondary perfusion was performed for 8–15 min using collagenase solution (Wakom, Japan), which contained NaCl (136.89 mM, Wako, Japan), KCl (5.36 mM, Wako, Japan), NaH$_2$PO$_4$/2H$_2$O (0.38 mM, Wako, Japan), Na$_2$HPO$_4$/12H$_2$O (0.17 mM, Wako, Japan), CaCl$_2$ (5.06 nM, Wako, Japan), HEPES (10 mM, DOJINDO, Japan), glucose (5 mM, Wako, Japan), NaHCO$_3$ (4.16 mM, Wako, Japan), phenol red (0.017 mM, Sigma-Aldrich Japan, Japan), and soybean trypsin inhibitor (0.0025 mM, Sigma-Aldrich Japan, Japan). After the secondary perfusion, the liver was split into multiple tissues, and the liver digestion was corrected through mesh filtration (Kawamoto, Japan) in 2% FBS MEM. To obtain the hepatocytes, we washed the digestion using MEM and corrected the pack with two centrifugations at 50×$g$. For purification of LSECs, we used the supernatant that was removed hepatocytes and performed MACS using anti-CD45 microbeads (130-109-682, Miltenyi, Germany), anti-PE microbeads (130-105-639, Miltenyi, Germany), and anti-CD146 PE-conjugated antibody (130-111-207, Miltenyi, Germany). We used the fraction of CD45− and CD146+ as LSECs.

**Cell culture and shear stress loading**. LSECs and HUVECs were cultured with the endothelial cell growth medium BulletKit for maintenance (ECM; CC-3124, Lonza, Switzerland). For the shear stress loading experiment, these cells were seeded onto a microslide I 0.4 Luer (id80176, ibidi, USA) coated with collagen 1A. We perfused endothelial cell basal medium (EBM; CC-3121, Lonza, Switzerland) supplemented with 0.5% BSA and an antibiotic-antimycotic mixed stock solution (09366-44, Nacalai Tesque, Japan) to load shear stress using a peristaltic pump. For inhibition of GPCRs, we precultured the cells with ECM supplemented with 0.25 μg/mL suramin sodium (193-10611, Wako, Japan) an hour before shear stress loading.

**Analysis of sinusoidal blood flow**. Animals were incised at the abdomen to expose the surface of the liver and placed on a plate made of thin glass to observe the liver surface at a wavelength of 488 nm under 1.5% isoflurane inhalation anaesthesia. We obtained time-lapse images to observe the erythrocytes moving at the liver surface using a high-speed imaging system constructed of an EM-CCD camera ImagEM X2 and high-speed recording software (C9100-23B, Hamamatsu Photonics, Japan). We tracked erythrocytes in every frame and measured the moving distance over ten frames using Imaris software.

**Calcium imaging**. The medium was replaced with ECM supplemented with 10 μM Indo-1 AM (I006, Dojindo, Japan) and 0.02% Pluronic F-127 (59005, Biotium, USA) for 50 min to incorporate the dye and ECM for 20 min to de-esterize the intracellular dye. We obtained time-lapse images using a two-photon laser microscope (LSM880 NLO, Carl Zeiss, Germany) by perfusing HBSS without calcium (17461-05, Nacalai Tesque, Japan) under a 20-cm hydraulic head distance to obtain a non-pulsatile flow of 11 mL/min. In the suramin group, we used these media supplemented with 0.25 μg/mL suramin sodium.

**Bypass construction and partial hepatectomy**. Rats were incised at the abdomen transversely, and the infrahepatic inferior vena cava was half-clamped to anastomose the vessel graft that was obtained from the other rat portal vein. After anastomosing, we clamped the portal vein and anastomosed it to the other end of the graft. Before opening the clamp, we stenosed the portal vein at the hepatic hilar section using a plastic tubular band made from 20G Surflo (Terumo, Japan) and opened the clamp. Continuously, we resected the left and median lobes of the liver for 2/3 partial hepatectomy according to the Higgins and Anderson method.

**s-OLT model**. Donor rats at 5 weeks of age were incised at the abdomen transversely, and the resected ligament was connected to the liver and ligated to the left diaphragmatic vein, portal branchings, right renal vein, and right adrenal vein under 2% isoflurane. The hepatic artery from the common hepatic artery to the proper hepatic artery was preserved, and branches such as the gastric artery and splenic artery were ligated. After a tail vein injection of 100 U heparin sodium (NIPRO, Japan), a 22G catheter (Terumo, Japan) was inserted into the bile duct as a stent. We clamped the thoracic aorta and flushed out the blood by perfusing saline from the abdominal aorta before liver donation. A plastic cuff of 20G catheter (Terumo, Japan) for the portal vein and 12G catheter (Angiocath, BD, USA) for the infrahepatic inferior vena cava (IHIVC) was attached to each donor's vessel carefully and with full dilation. To reconstruct the hepatic artery, we preserved the aortic wall around the branching to use it as a Currel's patch. We anaesthetized the recipient rat at 15 weeks using 1.5–2% isoflurane and removed the liver after clamping the portal vein, IHIVC, and suprahepatic inferior vena cava (SHIVC) and ligating the proper hepatic artery. We anastomosed the SHIVC manually by reefing the recipient's vascular end and inserted the portal cuff into the recipient portal vein. After reflowing to the transplanted liver, the IHIVC cuff was inserted into the recipient's vessel for subsequent reconstruction of the hepatic artery by anastomosing to the abdominal aorta and allowing blood to flow again. The bile ducts were connected to each other by the stent, and several clots in the abdomen were washed out using warm saline before abdominal closure.

**PCR**. RNA was isolated from hepatocytes and LSECs using the RNeasy Plus Mini Kit (Qiagen) and was reverse transcribed using SuperScript VILO (Thermo Fisher, USA) to obtain cDNA. Real-time PCR was performed on an Applied Biosystems QuantStudio 12K Flex (Thermo Fisher Scientific) using SYBR Premix Ex Taq II (TaKaRa Bio, Japan) or a TaqMan Gene Expression Assay system (Thermo Fisher, USA). The amount of expression of each target gene was normalized to the expression level of GAPDH. The primer pairs used for real-time PCR are listed in Supplementary Table 1.

**Western blotting**. The cells were washed with ice-cold PBS, and the proteins were extracted using RIPA buffer supplemented with protease and phosphatase inhibitor cocktails (5892970001 and 4906837001, Roche, Switzerland). The lysates were centrifuged at 13,000 rpm for 5 min, and supernatants were quantified using Bradford reagent. Western blotting was performed using p44/42 MAPK antibody (4695S, CST, USA), phospho-p44/42 MAPK antibody (4370S, CST, USA) and beta actin antibody (ab8226, Abcam, UK), and western blot images were analysed with ImageQuant LAS 3000 (GE Healthcare, USA).

**Quantification of cytokines**. Culture medium and serum were quantified using an ELISA kit for rTGFβ1 (MB100B, R&D Systems, USA) according to the manufacturer's protocol.

**Statistics and reproducibility**. Statistical significance was determined based on Student's $t$-test with two-tailed test. We repeated same experiments at least three times. Each sample size are indicated in the figure legend.

**Reporting summary**. Further information on research design is available in the Nature Research Reporting Summary linked to this article.

## Data availability
The datasets generated during and analyzed during the current study, including Supplementary data 1, are available from the corresponding author, T.T., on reasonable request.

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

## Acknowledgements
We thank lab members in RIKEN BDR, especially Ms. Y. Tamai, Ms. T. Iga, Ms. Y. Morioka, Ms. M. Takase, and Y. Higuchi, and we thank the animal facilities of BDR for technical assistance. We would like to thank Organ Technologies Inc. for funding support.

## Author contributions
T.T. and J.I. designed the research plan, J.I., A.I., J.K., Y.S., K.T., and A.K. performed the experiments; J.I., M.T., M.O, and T.T. discussed the results; J.I., A.I., J.K., M.K., and M.T. analysed the data; and T.T., J.I., and M.T. wrote the manuscript.

## Competing interests
The authors declare the following competing interests: T.T. is a supreme technical advisor at Organ Technologies Inc. This work was partially performed under the condition of an Invention Agreement between Tokyo University of Science and Organ Technologies Inc. The remaining authors declare no competing interests.
