## [Peer Review File · Communications Biology]

Reviewers' comments:

Reviewer #1 (Remarks to the Author):

The authors present very interesting results which document the essential role played by the sinusoidal endothelial network of the liver in initiation and regulation of liver regeneration after partial hepatectomy. The analysis presented is unique in the literature of liver regeneration and provides insightful perspectives in the overall interplay between liver regenerative growth and development of the new sinusoidal network towards the completion of regeneration. The mathematical analysis of the structures of the sinusoids and the connective vessels is also novel and the findings illustrate the different reactive changes between the two blood vessel systems.

The authors should address some small issues, as follows:

1. Though the authors do explain what they mean by "organ hypertrophy", nonetheless, the term hypertrophy is now widely used to indicate cellular enlargement (hypertrophy) versus cell proliferation (hyperplasia). The term "liver regeneration" should be used in this manuscript, to avoid confusion by readers. It is the term widely used to define and describe what the authors call "liver hypertrophy".
2. The authors should mention that enlargement of a baboon liver to that of human liver size also occurred following transplantation of the baboon liver to a human, by Thomas Starzl (PMID 8029900).
3. Previous studies described the rapid change of hepatocyte membrane potential after partial hepatectomy (Wondergem R and Harder DR, *J. of Cellular Physiology*, 102:193-197, 1980, Shang et al., *Hepatology*. 1996. PMID: 8617436). This suggests that the changes described by the authors, occurring in endothelial cells, in some way rapidly translate to electrochemical changes in the hepatocytes. Stellate cells, mimicking many neuron properties but not descended from neural crest, reside between endothelial cells and hepatocytes. The role of these cells as potential mediators between endothelial cells and hepatocytes needs to be mentioned in the Discussion.
4. Angiogenesis during liver regeneration is very complex process, involving proliferation of endogenous LSECs and also migration of endothelial cell progenitors from the bone marrow. The authors have accurately tabulated the overall changes occurring in the sinusoidal vasculature during liver regeneration, but it is also worth pointing out that many cellular elements are involved in the process.
5. It is not clear, from the authors description, whether the entire portal vein or only a branch of it was diverted to the inferior vena cava. The authors should clarify that point.
6. There are many signaling pathways involved in liver regeneration besides HGF, EGF and TGF-beta. The authors briefly mention this. A slight more coverage should be attempted (see Liver regeneration: biological and pathological mechanisms and implications, *Nat Rev Gastroenterol Hepatol*. 2020 Aug 6. doi: 10.1038/s41575-020-0342-4, PMID: 32764740). (Principles of Liver Regeneration and Growth Homeostasis, *Compr Physiol*. 2013 Jan;3(1):485-513. doi: 10.1002/cphy.c120014. PMID: 23720294).

Reviewer #2 (Remarks to the Author):

The authors of "Mechanical homeostasis of liver sinusoidal networks is involved in the initiation and termination of liver hypertrophy" present a technically and visually compelling paper regarding the role of sinusoidal endothelial cells regulating liver size. They perform a number of technically challenging surgical techniques to increase/decrease blood flow rate, resulting in changes in liver size that they attribute to mechanical tension upon endothelial cells which lead to paracrine changes to stimulate hepatocyte proliferation.

Although visually and technically impressive, the paper does not obviously conceptually advance the field as endothelial cells for many years have been shown to play a key role in regulating organ size. The papers by Bi-Sen Ding and Raffi Sahin in early 2010 (as cited by the the authors) nicely showed some of the molecular mechanisms that endothelial cells regulate liver size.

Coupling the surgical and visualization techniques the authors present with in vivo genetic or chemical modification of endothelial cells could conceptually advance the field. At the very least, transcriptionally or proteomically profiling the endothelial cells before and after their surgical manipulations would give more potentially novel insight into the changes that then lead to paracrine signaling to stimulate hepatocyte/HPC proliferation. Presentation of the work regarding TGF β and EGFR signaling although interesting and consistent with the current literature, does not meaningfully advance the field. Some of the changes transmitted by endothelial cells although presumed to be due to paracrine factors, may also be due to direct changes in cell tension, which could lead to proliferation. This too would be a novel insight into this field.

Directly answering a new mechanism of liver growth would make this a very compelling paper for the journal.

Response to Reviewer #1

We have studied your comments carefully and found that you understood the value and significance of our study in this field. We are grateful for your evaluation and valuable suggestions for our manuscript. Our specific responses are listed below:

General Comments:

The authors present very interesting results which document the essential role played by the sinusoidal endothelial network of the liver in initiation and regulation of liver regeneration after partial hepatectomy. The analysis presented is unique in the literature of liver regeneration and provides insightful perspectives in the overall interplay between liver regenerative growth and development of the new sinusoidal network towards the completion of regeneration. The mathematical analysis of the structures of the sinusoids and the connective vessels is also novel and the findings illustrate the different reactive changes between the two blood vessel systems.

Our Response: We are grateful for your understanding of our findings and highly evaluation of our manuscript.

Specific Comments:

1. Though the authors do explain what they mean by “organ hypertrophy”, nonetheless, the term hypertrophy is now widely used to indicate cellular enlargement (hypertrophy) versus cell proliferation (hyperplasia). The term “liver regeneration” should be used in this manuscript, to avoid confusion by readers. It is the term widely used to define and describe what the authors call “liver hypertrophy”.

Our Response: Thank you for kindly pointing this out. According to your indication, we have replaced the term “liver hypertrophy” to “liver regeneration” for partial hepatectomy and bypass model and to “liver growth” for s-OLT model.

2. The authors should mention that enlargement of a baboon liver to that of human liver size also occurred following transplantation of the baboon liver to a human, by Thomas Starzl (PMID 8029900).

Our Response: We are grateful for your advice. As you suggested, we have mentioned about the enlargement of baboon liver after transplantation to human in the Result Section.

3. Previous studies described the rapid change of hepatocyte membrane potential after partial hepatectomy (Wondergem R and Harder DR, J. of Cellular Physiology, 102:193-197, 1980, Shang et al., Hepatology. 1996. PMID: 8617436). This suggests that the changes described by the authors, occurring in endothelial cells, in some way rapidly translate to electrochemical changes in the hepatocytes. Stellate cells, mimicking many neuron properties but not descended from neural crest, reside between endothelial cells and hepatocytes. The role of these cells as potential mediators between endothelial cells and hepatocytes needs to be mentioned in the Discussion.

Our Response: We are grateful for your kindly suggestion to have roles of endothelial cells and stellate cells for liver regenerations triggered by shea stress. According to your valuable suggestions, we have revised the possibility of their role in the Discussion Section.

4. Angiogenesis during liver regeneration is very complex process, involving proliferation of endogenous LSECs and also migration of endothelial cell progenitors from the bone marrow. The authors have accurately tabulated the overall changes occurring in the sinusoidal vasculature during liver regeneration, but it s also worth pointing out that may cellular elements are involved in the process.

Our Response: According to the your comment, we have described that multiple cell types involves in liver regeneration in the Discussion Section.

5. It is not clear, from the authors description, whether the entire portal vein or only a branch of it was diverted to the inferior vena cava. The authors should clarify that point.

Our Response: Thank you for pointing this out. We have changed the description of by-pass model as below:

‘To this end, we developed a surgical rat model that can suppress the increased blood flow velocity in sinusoids by redirecting about 70% of blood flow of portal vein into the inferior vena cava by delegating the portal vein to reduce the cross-sectional area to 1/3 of the original (Fig. 3a, b)’

Adjusted so that the cross-sectional area is 1/3

6. There are many signaling pathways involved in liver regeneration besides HGF, EGF and TGF-beta. The authors briefly mention this. A slight more coverage should be attempted (see Liver regeneration: biological and pathological mechanisms and implications, Nat Rev Gastroenterol Hepatol. 2020 Aug 6. doi: 10.1038/s41575-020-0342-4, PMID: 32764740). (Principles of Liver Regeneration and Growth Homeostasis, Compr Physiol. 2013 Jan;3(1):485-513. doi: 10.1002/cphy.c120014.PMID: 23720294).

Our Response: We are grateful for your kindly suggestion. According to your comment, we have revised to incorporate the several signaling pathways, such as Tumour necrosis factor (TNF), vascular endothelial growth factor (VEGF), Hedgehog, Wnt/beta-catenin, involved in liver regeneration in the Introduction Section.

We hope that these changes meet with your approval. We greatly appreciate your comments, which provided a helpful perspective on our work.

Response to Reviewer #2

We are grateful for your evaluation and valuable suggestions for our manuscript. Our responses are listed below:

Comment 1:

The authors of "Mechanical homeostasis of liver sinusoidal networks is involved in the initiation and termination of liver hypertrophy" present a technically and visually compelling paper regarding the role of sinusoidal endothelial cells regulating liver size. They perform a number of technically challenging surgical techniques to increase/decrease blood flow rate, resulting in changes in liver size that they attribute to mechanical tension upon endothelial cells which lead to paracrine changes to stimulate hepatocyte proliferation.

Although visually and technically impressive, the paper does not obviously conceptually advance the field as endothelial cells for many years have been shown to play a key role in regulating organ size. The papers by Bi-Sen Ding and Raffi Sahin in early 2010 (as cited by the the authors) nicely showed some of the molecular mechanisms that endothelial cells regulate liver size.

Our Response: First of all, we are grateful for highly evaluation of visually and technically in our work. As reviewer pointed out, we have understood and described in the Introduction Section that the importance of vascular endothelium has already been reported. In the current study, we utilized the liver as a model system; the mass/volume of the hepatic lobule is essential for liver function but not organ morphology, in contrast to other organs with specific shapes and morphologies. This system was used for analyses of detailed three-dimensional sinusoidal structure and liver regeneration as an organ regeneration model. We reveal that mechanical homeostasis in blood vessels is one of the keys to detecting the state of the entire organ and converts this information into a cytokine network that triggers the initiation and termination of cell proliferation, which is vital for controlling organ size.

What is new in this work is that the vascular endothelium monitors the condition of the entire liver via shear stress, which is vital for perform liver regeneration. Our finding is one of the answers to the long-standing question of how the condition of the entire organ is monitored and how cells initiate and/or terminate to proliferate and converge at the right time in organ development and regeneration. Then, this shear stress via suramin-sensitive G-protein-coupled receptors (GPCRs) and Ca^{2+} -influx in

both HUVEC and liver-derived sinusoidal endothelial cells, would regulate the known cytokine networks and the proliferation of liver cells. We believe that our findings are a sufficiently novel concept that applies not only to liver regeneration but also to development and regeneration of other organs.

Comment 2:

Coupling the surgical and visualization techniques the authors present with in vivo genetic or chemical modification of endothelial cells could conceptually advance the field. At the very least, transcriptionally or proteomically profiling the endothelial cells before and after their surgical manipulations would give more potentially novel insight into the changes that then lead to paracrine signaling to stimulate hepatocyte/HPC proliferation. Presentation of the work regarding TGF β and EGFR signaling although interesting and consistent with the current literature, does not meaningfully advance the field. Some of the changes transmitted by endothelial cells although presumed to be due to paracrine factors, may also be due to direct changes in cell tension, which could lead to proliferation. This too would be a novel insight into this field. Directly answering a new mechanism of liver growth would make this a very compelling paper for the journal.

Our Response: We are grateful for your evaluation that coupling the surgical and visualization techniques the authors present with in vivo genetic or chemical modification of endothelial cells could conceptually advance the field. We also thank for your constructive suggestions. We totally agree with your suggestions including the transcriptional and proteomics analyses after the surgical treatment. However, the focus of this study is how cells monitor the condition of the entire organ and how cells know the timing to initiate and terminate the proliferation and the detailed mechanism of liver regeneration is out of focus at this time. In this work, our findings strongly suggest that the balance of suramin-sensitive mechanical homeostasis and Ca²⁺-ion channel on the surface of endothelial cells in the sinusoidal network in the liver plays essential roles in liver regeneration, i.e. compensative hypertrophy prior to cytokine networks such as TGF- β and EGFR signaling. As we mentioned in the Discussion section, the points, which you suggested, are essential to fully understand the mechanism of liver regeneration and need to be elucidated in the next future study.

We hope that these changes meet with your approval. We greatly appreciate your comments, which provided a helpful perspective on our work.

REVIEWERS' COMMENTS:

Reviewer #1 (Remarks to the Author):

No comments necessary.

Reviewer #2 (Remarks to the Author):

The authors of, "Mechanical homeostasis of liver sinusoidal networks is involved in the initiation and termination of liver regeneration" present a minimally revised manuscript from their previous submission. While I continue to find their data visually compelling, the idea that sinusoidal endothelial cells and shear stress across the cell as an initiator of liver regeneration is not particularly novel. Other investigators (PMID: 17047332, 18395841, 20148099) have used similar techniques, although not as elegantly as the current study. While the authors debate if molecular linkage to novel mechanisms is within the scope of this study, the authors should make a real effort to address prior work and how this study is an improvement upon those and other studies in the Discussion.

The authors suggest that suramin-sensitive GPCRs in endothelial cells initiate hepatocyte replication, direct evidence of this in vivo would greatly strengthen the paper. Decoupling shear stress sensing in LSECs and its role in hepatocyte replication will make for a more consistent story. Their in vitro work using HUVECs and cultured LSECs in a perfusion chamber (Figure 4) suggested that this would be experimentally followed by a similar monitoring system in vivo, but this did not occur. The authors instead present a small for liver size model leaving the reader to consider if partial hepatectomy in vivo does not lead to calcium influxes or ERK phosphorylation.

Response to Reviewer #1

Comment:

No comments necessary.

Our Response:

We are grateful for your valuable and instructive comments and suggestions to improve our manuscript.

Response to Reviewer #2

We would like to thank reviewer to for evaluate our manuscript and give important suggestions. Our responses are listed below:

Specific Comment1:

The authors of,"Mechanical homeostasis of liver sinusoidal networks is involved in the initiation and termination of liver regeneration" present a minimally revised manuscript from their previous submission. While I continue to find their data visually compelling, the idea that sinusoidal endothelial cells and shear stress across the cell as an initiator of liver regeneration is not particularly novel. Other investigators (PMID: 17047332, 18395841, 20148099) have used similar techniques, although not as elegantly as the current study. While the authors debate if molecular linkage to novel mechanisms is within the scope of this study, the authors should make a real effort to address prior work and how this study is an improvement upon those and other studies in the Discussion.

Our Response:

We appreciate you to pointing this out. We totally agree with you that the concept of a correlation between shear stress and regeneration is not novel. However, previous studies, including researches you mentioned (PMID: 17047332, 18395841, 20148099), suggests this concept but did not precisely examined the causal relationship between shear stress and liver regeneration. This is partially because of that various factors are involved in regeneration process makes it difficult to examine the causal relationship between one factor and regeneration *in vivo*.

In current study, to overcome this limitation, we used *in vitro* model to simplify the factors involved in the liver regeneration and demonstrated that the production of

TGFb1, which is one of the key trigger for liver regeneration, from vascular endothelial cells decreases in response to stress. Our result is the first direct evidence that suggest the causal relationship between shear stress and liver regeneration.

As you pointed out, it is necessary to confirm whether the mechanism discovered *in vitro* actually works *in vivo*. Nevertheless, we believe that our finding added new insight into the liver regeneration and improve the concept of a correlation between shear stress and regeneration. According to your suggestion, we mentioned previous studies and discussed this point in the Discussion.

Specific Comment2:

The authors suggest that suramin-sensitive GPCRs in endothelial cells initiate hepatocyte replication, direct evidence of this in vivo would greatly strengthen the paper. Decoupling shear stress sensing in LSECs and its role in hepatocyte replication will make for a more consistent story. Their in vitro work using HUVECs and cultured LSECs in a perfusion chamber (Figure 4) suggested that this would be experimentally followed by a similar monitoring system in vivo, but this did not occur. The authors instead present a small for liver size model leaving the reader to consider if partial hepatectomy in vivo does not lead to calcium influxes or ERK phosphorylation.

Our Response:

We are grateful for your suggestion to make our paper stronger. As you suggested, we have tried to confirm Ca^{2+} influx and ERK phosphorylation *in vivo* but it did not work well because of the experimental limitations as below.

To examine if suramin prevent liver regeneration, we administrated suramin at a concentration that does not harm rodents' health. However, we did not significant change between control and suramin administrated group. This may be because of the concentration of suramin is not sufficient to affect liver regeneration. Because GPCR signaling have essential role in multiple biological process *in vivo*, we could not increase the concentration of suramin and examined above point.

The change in Ca^{2+} influx occurs within few second after suramin administration, which can only be detected by intravital imaging. However, no method has been established so far for Ca^{2+} influx imaging of sinusoidal endothelium. Therefore, we could not examine whether the change in Ca^{2+} influx occurs also *in vivo*.

In current study, performed collagenase perfusion magnetic activated cell sorting (MACS) to obtain purified LSECs from the liver. Because this process takes few hours, isolated LSECs do not reflect the state immediately after treatment.

We totally agree with you that *in vivo* functional confirmation is required to fully understand the mechanism how entire liver monitor their situation. However, our results still give important insight possible monitoring mechanism that LECs sense changes in blood flow and alter the production of cytokines involved in liver regeneration. We discussed above limitations in the Discussion.

We hope that these changes meet with your approval. We greatly appreciate your helpful comments.